# Time without PSA recurrence after radical prostatectomy as a predictor of future biochemical recurrence, metastatic disease and prostate cancer death: a prospective Scandinavian cohort study

Mats Steinholtz Ahlberg ![ORCID],[1] Hans Garmo,[1] Hans-Olov Adami,[2,3] Ove Andrén,[4] Jan-Erik Johansson,[4] Gunnar Steineck,[5] Lars Holmberg ![ORCID],[6] Anna Bill-Axelson[1]

For numbered affiliations see end of article.

**Correspondence to**
Dr Mats Steinholtz Ahlberg; mats.ahlberg@surgsci.uu.se

## ABSTRACT

**Objective** Although surveillance after radical prostatectomy routinely includes repeated prostate specific antigen (PSA)-testing for many years, biochemical recurrence often occurs without further clinical progression. We therefore hypothesised that follow-up can be shortened for many patients without increasing the risk of prostate cancer death. We investigated the long-term probabilities of PSA recurrence, metastases and prostate cancer death in patients without biochemical recurrence five and 10 years after radical prostatectomy.

**Design** Prospective cohort study. Stratification by Gleason score (≤3+4=7 or ≥4+3=7), pathological tumour stage (pT2 or ≥pT3) and negative or positive surgical margins.

**Setting** Between 1989 and 1998, 14 urological centres in Scandinavia randomised patients to the Scandinavian Prostate Cancer Group study number 4 (SPCG-4) trial.

**Participation** All 306 patients from the SPCG-4 trial who underwent radical prostatectomy within 1 year from inclusion were eligible. Four patients were excluded due to surgery-related death (n=1) or salvage radiotherapy or hormonal treatment within 6 weeks from surgery (n=3).

**Primary outcome measures** Cumulative incidences and absolute differences in metastatic disease and prostate cancer death.

**Results** We analysed 302 patients with complete follow-up during a median of 24 years. Median preoperative PSA was 9.8 ng/mL and median age was 65 years. For patients without biochemical recurrence 5 years after radical prostatectomy the 20-year probability of biochemical recurrence was 25% among men with Gleason score ≤3+4=7 and 57% among men with Gleason score ≥4+3=7; the probabilities for metastases were 0.8% and 17%; and for prostate cancer death 0.8% and 12%, respectively. The long-term probabilities were higher for pT ≥3 versus pT2 and for positive versus negative surgical margins. Limitations include small size of the cohort.

**Conclusion** Many patients with favourable histopathology without biochemical recurrence 5 years after radical prostatectomy could stop follow-up earlier than 10 years after radical prostatectomy.

## STRENGTHS AND LIMITATIONS OF THIS STUDY

⇒ The long and complete prospective follow-up of relevant endpoints strengthen the validity of our results.
⇒ The fact that an independent committee determined cause of death increases the reliability of that endpoint.
⇒ The small size of the cohort is a limitation.

## INTRODUCTION

Because lethal and prostate specific antigen (PSA) detected non-lethal prostate cancer cannot be reliably separated, millions of men who have undergone radical prostatectomy are unnecessarily followed for many years with repeated PSA tests in order to detect a biochemical recurrence. The American Urological Association guidelines recommend at least 10 years follow-up[1] while the European Association of Urology guidelines have no stopping recommendation even if PSA remains undetectable.[2] Thus, the potential benefit of follow-up needs to be weighed against the physical and psychological burden for the patient as well as the use of healthcare resources. However, it is not clear whether it is possible to shorten follow-up after radical prostatectomy for some patients, without increasing the risk of prostate cancer death.

After a radical prostatectomy, the risk of biochemical recurrence is highest during the first 2 years but still remains after 10 years.[3 4] High preoperative PSA, high Gleason score, high pathological tumour (pT)-stage and positive surgical margins are associated with increased risk of biochemical recurrence.[5] In men with biochemical recurrence, high Gleason score and short PSA-doubling time are associated with increased risk of metastases and prostate cancer death while there

are conflicting results regarding of pT-stage, surgical margins and time to biochemical recurrence.[4 6–11]

We investigated the probabilities of future biochemical recurrence, metastases and death from prostate cancer conditioned on 5 and 10 years without biochemical recurrence for men with Gleason score ≤3+4=7 or ≥4+3=7, pT2 or ≥pT3, and negative or positive surgical margins in the prostatectomy specimen. We hypothesised that men with no biochemical recurrence and favourable histopathology (Gleason score ≤3+4=7, pT2 or negative surgical margins) could stop follow-up earlier than 10 years after radical prostatectomy without increasing the probability of metastases or prostate cancer death.

## METHODS
### Patients
We included all patients from the Scandinavian Prostate Cancer Group study number 4 (SPCG-4) who underwent radical prostatectomy within 1 year from randomisation (1989–1998). All data were derived from the SPCG-4 database. Details of the trial design has been previously published.[12] Briefly, a total of 695 patients in 14 urological centres in Scandinavia were randomised to either watchful waiting or radical prostatectomy. All men gave oral consent to participate in the SPCG-4 study before randomisation. A total of 306 patients (289 randomised to radical prostatectomy and 17 randomised to watchful waiting) underwent radical prostatectomy, without evidence of lymph node invasion at frozen section, within 1 year from randomisation of whom 302 were included in the study (figure 1). No patient was lost to follow-up. About 75% of the patients in SPCG-4 had a palpable, clinically detected prostate cancer (cT2) while the remaining were detected by PSA testing or incidentally after transurethral resection of the prostate. The initial protocol for SPCG-4 promoted hormonal treatment only after onset of metastases.[13] After introduction of Bicalutamide in 2003, urologists could use antiandrogen therapy as desired according to an amendment to the protocol.[14] An independent committee determined cause of death in all deceased patients.

We defined biochemical recurrence after radical prostatectomy as the second of two separate PSA values ≥0.2 ng/mL or one single PSA value >0.4 ng/mL. We also defined biochemical recurrence as initiation of hormonal treatment (two patients) or salvage radiotherapy (seven patients) if occurring before a PSA definition was reached. For all definitions at least 6 weeks had to pass after radical prostatectomy. We defined positive surgical margin as any amount of prostate cancer in the resection margin, and extra-prostatic extension of prostate cancer (pT3) as any tumour growth outside the prostate capsule. Due to similarities in long-term risk of metastases and prostate cancer death between Gleason score ≤3+3=6 and 3+4=7

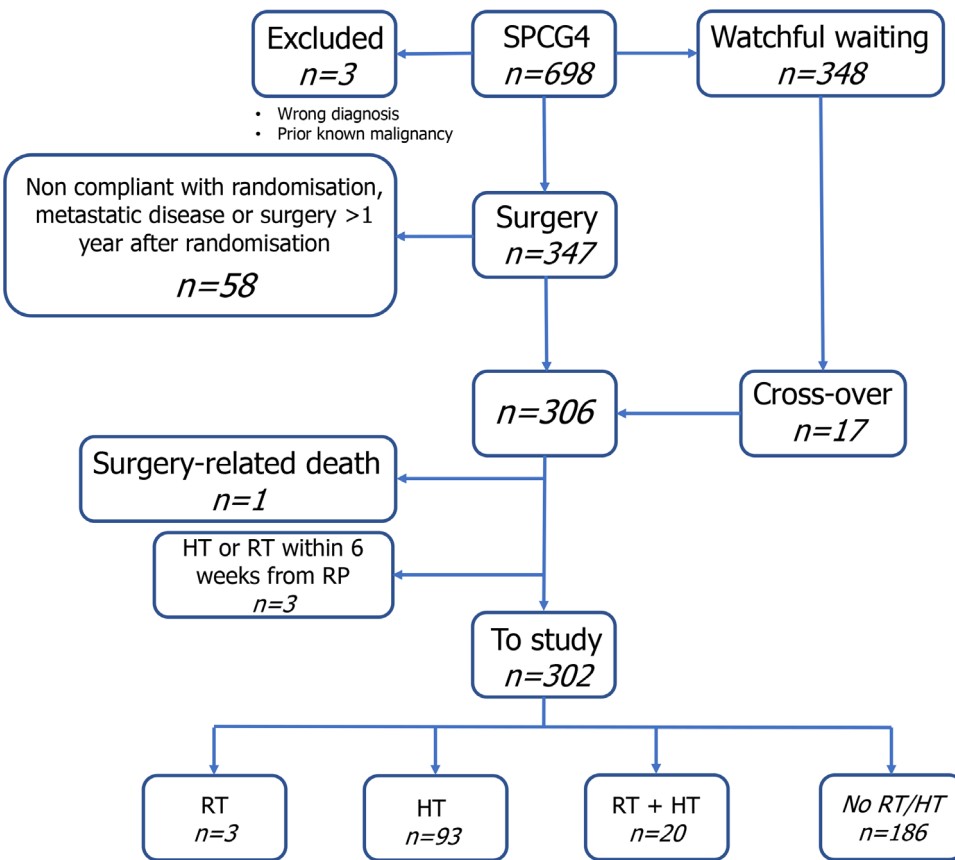

**Figure 1** Patient selection and complementary treatment after radical prostatectomy. HT, hormonal treatment; RP, radical prostatectomy; RT, radiotherapy; SPCG4, Scandinavian Prostate Cancer Group study nr. 4.

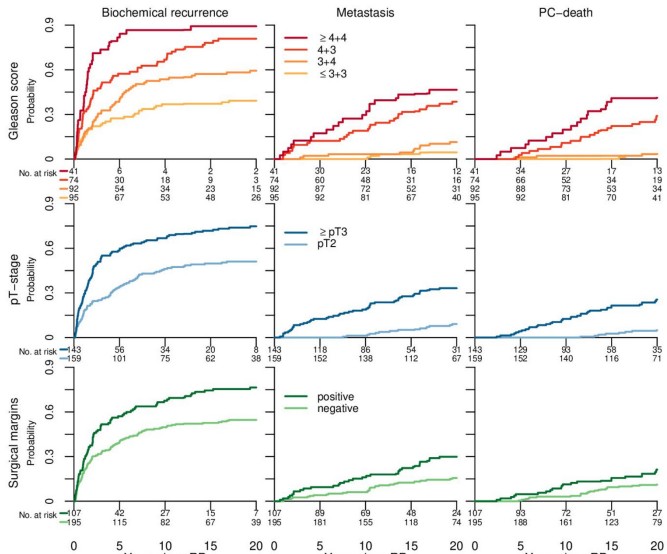

**Figure 2** Cumulative incidence of biochemical recurrence, metastasis and death from prostate cancer after radical prostatectomy. BCR, biochemical recurrence; PC, prostate cancer; pT, pathological T-stage; RP, radical prostatectomy.

(figure 2) we grouped patients into a favourable Gleason score-group (Gleason score ≤3+4=7) and a non-favourable Gleason score-group (Gleason score ≥4+3=7).[15] We also grouped patients by pT-stage (pT2 or ≥pT3) and negative or positive surgical margins. Preoperative PSA refers to the last PSA before surgery.

## Statistical analyses
We considered preoperative PSA and age as continuous variables and Gleason score, pT-stage and surgical margins as categorical variables. We present not normally distributed data as median and IQR.

We estimated the long-term probability of future biochemical recurrence, metastases and prostate cancer death in men without biochemical recurrence, as cumulative incidence proportions conditioned on time after radical prostatectomy without biochemical recurrence.[16] We considered death from other causes as a competing risk and censored at last day of follow-up. Follow-up was quantified using reverse Kaplan-Meier estimate of potential follow-up.[17 18]

Missing data were handled by imputation using multiple imputation by changed equations.[19] We estimated variance according to Rubin's rule[20] and performed a sensitivity analysis of one single PSA >0.6 ng/mL versus >0.4 ng/mL as one of the definitions of biochemical recurrence.

## Patient and public involvement
Patient experiences and priorities are important knowledge that may influence the definition of research questions. Because we used existing data from the SPCG-4 study database, patients and the public were not involved in the design or conduct of the study, or in the interpretation of the study results.

**Table 1** Baseline characteristics of patients included in the study

| n | 302 |
|---|---|
| Age, years | 65 (60–68) |
| Preoperative PSA, ng/mL | 9.8 (5.9–16.2) |
| Missing PSA | 7 (2.3) |
| pGS | |
| ≤3+3 | 88 (29.1) |
| 3+4 | 86 (28.5) |
| 4+3 | 69 (22.8) |
| ≥4+4 | 37 (12.3) |
| Missing | 22 (7.3) |
| pT | |
| pT2 | 146 (48.3) |
| ≥pT3 | 135 (44.7) |
| Missing | 21 (7.0) |
| Surgical margins | |
| Negative margin | 183 (60.6) |
| Positive margin | 97 (32.1) |
| Missing | 22 (7.3) |

Continuous data presented as median and IQR. Categorical data presented in number and per cent.
pGS, pathological Gleason score; PSA, prostate specific antigen; pT, pathological T-stage.

## RESULTS
### Descriptive characteristics
In all, 302 men who underwent radical prostatectomy within 1 year from randomisation fulfilled the criteria for this study (figure 1). Median age at inclusion was 65 years, median (IQR) follow-up time was 24 (21–26) years and median preoperative PSA 9.8 ng/mL. Fifty-eight per cent of the radical prostatectomy specimens contained Gleason score ≤3+4=7, 48% were pT2 and 61% had negative surgical margins (table 1).

During follow-up 218 men died. We documented biochemical recurrence in 190 men (63%), metastases in 63 men (21%) and prostate cancer death in 49 men (16%). Cumulative incidences for subgroups are shown in figure 2. Results of data analyses before and after imputation did not change the main result. Nor did sensitivity analysis of one single PSA >0.6 ng/mL versus >0.4 ng/mL—as one of the definitions of biochemical recurrence—change the results.

### Outcomes 20 years after radical prostatectomy conditioned on 5 years without biochemical recurrence
The probability of an event 20 years after radical prostatectomy conditioned on time after surgery without biochemical recurrence is illustrated in figure 3. Table 2 shows the estimates at 10, 15 and 20 years after radical prostatectomy; these results are based on the same analysis as shown in figure 3. Absolute differences (95% CI)

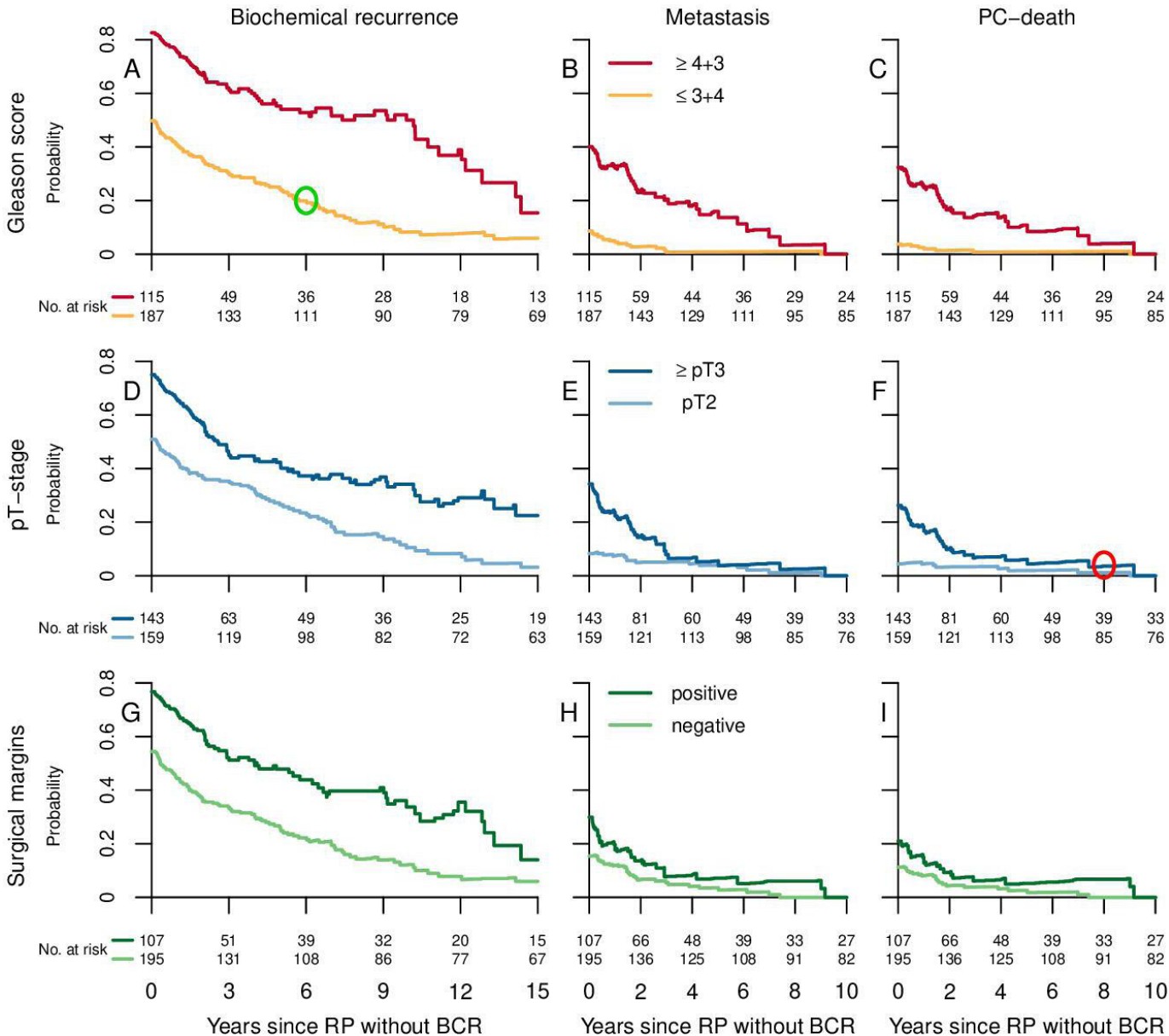

**Figure 3** 20-year probability of event conditioned on time after radical prostatectomy without biochemical recurrence. Y-axis represents the probability of an event (biochemical recurrence, metastasis and PC-death) for the different subgroups within 20 years after radical prostatectomy; X-axis represents time after radical prostatectomy without biochemical recurrence. The green circle in (A) exemplifies the probability to experience a biochemical recurrence within 20 years from radical prostatectomy for a patient with GS ≤3+4=7 who was free from biochemical recurrence 6 years after radical prostatectomy. The red circle in (F) exemplifies the probability to die from prostate cancer within 20 years from radical prostatectomy for a patient with ≥pT3 who was free from biochemical recurrence 8 years after radical prostatectomy. BCR, biochemical recurrence; GS, Gleason score; PC, prostate cancer; pT, pathological T-stage; RP, radical prostatectomy.

for Gleason score ≤3+4=7 versus ≥4+3=7; pT2 versus pT ≥3 and negative versus positive surgical margins, are shown in table 2. Twenty years after radical prostatectomy, the probabilities were 32% for biochemical recurrence, 4.5% for metastases and 3.3% for death from prostate cancer for all patients without biochemical recurrence 5 years after radical prostatectomy.

Among patients without biochemical recurrence 5 years after radical prostatectomy, the 20-year probability of biochemical recurrence was 25% among those with Gleason score ≤3+4=7 and 57% among those with Gleason

score ≥4+3=7; the corresponding probabilities for metastases were 0.8% and 17%; and for prostate cancer death 0.8% and 12%, respectively. Among men with Gleason score ≤3+4=7 without biochemical recurrence 5 years after radical prostatectomy, 11 out of 111 men later received hormonal treatment for prostate cancer of whom two also underwent salvage radiotherapy.

The 20-year probabilities of all events were numerically higher for pT ≥3 versus pT2 and for positive versus negative surgical margins even though the differences were not statistically significant for other than the probabilities

**Table 2** Probability (%) and absolute difference (% (95% CI)) of future biochemical recurrence, metastasis and prostate cancer death for men without biochemical recurrence 5 years after radical prostatectomy

| Probability (%) and absolute difference (% (95% CI)) of future biochemical recurrence for men without biochemical recurrence 5 years after radical prostatectomy | | | |
|---|---|---|---|
| Time after radical prostatectomy | 10 years | 15 years | 20 years |
| GS ≤3+4 | 18.2 | 21.1 | 24.5 |
| GS ≥4+3 | 25.1 | 51.3 | 56.9 |
| *Absolute difference* | *6.9 (–9.1 to 22.9)* | *30.1 (11.8 to 48.5)* | *32.4 (14.0 to 50.8)* |
| pT2 | 18.8 | 24.5 | 26.5 |
| pT3 | 21.6 | 34.5 | 42.2 |
| *Absolute difference* | *2.8 (–10.5 to 16.1)* | *10.0 (–5.2 to 25.2)* | *15.7 (–0.2 to 31.6)* |
| SM– | 17.4 | 22.4 | 25.9 |
| SM+ | 26.0 | 43.3 | 48.3 |
| *Absolute difference* | *8.6 (–6.5 to 23.7)* | *20.9 (3.8 to 37.9)* | *22.4 (4.9 to 39.8)* |
| **Probability (%) and absolute difference (% (95% CI)) of metastases for men without biochemical recurrence 5 years after radical prostatectomy** | | | |
| Time after radical prostatectomy | 10 years | 15 years | 20 years |
| GS ≤3+4 | 0.0 | 0.0 | 0.8 |
| GS ≥4+3 | 2.8 | 8.4 | 16.9 |
| *Absolute difference* | *2.8 (–2.7 to 8.3)* | *8.4 (–0.9 to 17.6)* | *16.1 (3.3 to 28.8)* |
| pT2 | 1.0 | 1.0 | 4.0 |
| pT3 | 0.0 | 3.6 | 5.4 |
| *Absolute difference* | *1.0 (–0.9 to 2.9)* | *2.6 (–2.7 to 7.9)* | *1.4 (–5.8 to 8.5)* |
| SM– | 0.9 | 2.6 | 3.6 |
| SM+ | 0.0 | 0.0 | 7.1 |
| *Absolute difference* | *0.9 (–0.8 to 2.6)* | *2.6 (–0.3 to 5.6)* | *3.5 (–5.1 to 12.2)* |
| **Probability (%) and absolute difference (% (95% CI)) of prostate cancer death for men without biochemical recurrence 5 years after radical prostatectomy** | | | |
| Time after radical prostatectomy | 10 years | 15 years | 20 years |
| GS ≤3+4 | 0.0 | 0.0 | 0.8 |
| GS ≥4+3 | 0.0 | 8.4 | 11.5 |
| *Absolute difference* | *0.0* | *8.4 (-0.9 to 17.6)* | *10.7 (-0.3 to 21.8)* |
| pT2 | 0.0 | 1.0 | 2.0 |
| pT3 | 0.0 | 3.6 | 6.0 |
| *Absolute difference* | *0.0* | *2.6 (-2.7 to 7.9)* | *4.0 (-3.3 to 11.3)* |
| SM– | 0.0 | 2.6 | 2.6 |
| SM+ | 0.0 | 0.0 | 5.2 |
| *Absolute difference* | *0.0* | *2.6 (–0.3 to 5.6)* | *2.6 (–5.2 to 10.4)* |

GS, Gleason score; pT, pathological T-stage; SM+, positive surgical margins; SM–, negative surgical margins.

of biochemical recurrence for patients with positive versus negative surgical margins. The probabilities of biochemical recurrence varied between 17% for men with negative surgical margins 10 years after radical prostatectomy and 48% for men with positive surgical margins 20 years after radical prostatectomy. The probabilities of death from prostate cancer resembled the probabilities of metastases but were considerably lower than the probabilities of biochemical recurrence (table 2).

For patients without biochemical recurrence, the 20-year probabilities of metastases and death from prostate cancer in the whole cohort decreased by at least two-thirds during the first 3 years after radical prostatectomy (from 21% to 6% for metastases and from 15% to 5% for prostate cancer death) after which the decrease flattened out. We saw the same pattern, a decrease in probability of about 50% or more in the first 3 years, in all subgroups except in men with pT2 (figure 3).

### Outcomes 20 years after radical prostatectomy conditioned on 10 years without biochemical recurrence

The probability of biochemical recurrence 20 years after radical prostatectomy for all patients without biochemical recurrence 10 years after radical prostatectomy was 17%. Among patients without biochemical recurrence 10 years after radical prostatectomy none was diagnosed with metastases or died from prostate cancer (figure 3). The 20-year probabilities (95% CI of absolute differences) of future biochemical recurrence were 9% for Gleason score ≤3+4=7 and 52% (21–66) for Gleason score ≥4+3=7; 10% for pT2 and 34% (6–42) for ≥pT3; and 12% for those with a negative and 34% (3–42) for those with positive surgical margins. Among men with Gleason score ≥4+3=7 without biochemical recurrence 10 years after radical prostatectomy, 3 out of 20 received hormonal treatment and none underwent salvage radiotherapy.

## DISCUSSION
### Principal findings
Following radical prostatectomy, patients with Gleason score ≤3+4=7 without biochemical recurrence 5 years after radical prostatectomy had low risk of metastases and prostate cancer death independent of pT-stage and surgical margins. The risk of clinical progression decreased drastically in the first 3 years after radical prostatectomy and after 10 years without biochemical recurrence, no patient was diagnosed with metastases or died from prostate cancer.

### Comparisons with other studies
Loeb et al[21] presented evidence for low risk of cancer related morbidity and mortality within 20 years after radical prostatectomy if PSA was undetectable 10 years after radical prostatectomy. Ahove et al,[22] with 10 years of follow-up, showed that it is unlikely for patients with Gleason score 6 to develop late biochemical recurrence if PSA was undetectable 5 years after radical prostatectomy. In the present study with median follow-up of 24 years, men without biochemical recurrence 5 years after radical prostatectomy still had a rather high probability of future biochemical recurrence, while the probability of metastases and prostate cancer death varied. Gleason score was the strongest predictor of outcomes. Among men with Gleason score ≥4+3=7 the long-term probability of biochemical recurrence was about two times higher than for men with Gleason score ≤3+4=7 while the probabilities of metastasis and prostate cancer death were about 20 and 15 times higher, respectively. In all, 7 of 157 men with biochemical recurrence after more than 5 years were later diagnosed with metastatic disease and six of them died from prostate cancer. Only one of these men was in the favourable Gleason score-group.

Several other studies evaluated how time to biochemical recurrence predicts metastases and prostate cancer death after radical prostatectomy with somewhat conflicting results. Freedland et al and Cair et al found

that shorter time to biochemical recurrence was associated with higher prostate cancer specific mortality while Bolton et al was able to demonstrate this association in men with intermediate and high-risk prostate cancer but not in men with low-risk prostate cancer.[9 10 23] Pound found evidence for that shorter time to biochemical was associated with metastases but not increased mortality.[4] Boorjian et al, Zhou et al, Ward et al and Antonarakis et al, however, found no association between interval from radical prostatectomy to biochemical recurrence and oncological outcomes after adjusting for other clinicopathological features.[3 6–8 11] No previous study had as long follow-up as in our study. Our findings clearly support that a longer interval between surgery and biochemical recurrence predicts lower risk of metastases and prostate cancer death. Our findings of a rapid decrease in probability of metastases and death from prostate cancer the first 3 years after radical prostatectomy are supported by earlier studies where 2–3 years after radical prostatectomy appears to be a cut-off when the risk of clinical progression declines.[3 4]

Ten per cent of men in the favourable Gleason score-group without biochemical recurrence 5 years after radical prostatectomy eventually received hormonal treatment. Some recent data indicate that immediate androgen deprivation therapy after biochemical recurrence can improve overall survival for men with biochemical recurrence after radical prostatectomy.[24] Other data suggest that early hormonal treatment is unlikely to reduce the risk of clinical progression in patients with favourable histopathology.[25 26] In our study, 23 men underwent salvage radiotherapy, of which only two were in the favourable Gleason score-group without biochemical recurrence 5 years after radical prostatectomy. Salvage radiotherapy after biochemical recurrence can provide sustainable PSA response and a survival benefit in men with a PSA doubling time less than 6 months.[27] The timing of salvage radiotherapy is debated but early treatment is probably beneficial for patients with long life expectancy and non-favourable histopathology.[28 29] Thus, in our study men with Gleason score ≤3+4=7 without biochemical recurrence 5 years after radical prostatectomy, hormonal treatment or salvage radiotherapy is unlikely to substantially reduce risk of metastases or prostate cancer death.

We did not adjust for pT-stage or surgical margins when we analysed probabilities of future metastases and prostate cancer death in the two Gleason score-groups. Thus, in the favourable Gleason score-group some patients had ≥pT3 tumours and/or positive surgical margins. In our study also other favourable histopathological characteristics (pT2 and negative surgical margins) predicted low risk of clinical progression. These findings support shorter follow-up in men with favourable histopathological characteristics without biochemical recurrence at 5 years. However, some patients will undergo surgery at a young age with far more than 20 years of expected remaining lifetime. Our results do not support a shorter follow-up in

this patient group as their time at risk for metastases and prostate cancer death exceeds the follow-up of this study.

Biochemical recurrence after 10 years was fairly common. We found, however, no metastatic disease and no death from prostate cancer in men without biochemical recurrence 10 years after radical prostatectomy. Only three men in the non-favourable Gleason score-group without biochemical recurrence after 10 years received hormonal treatment. Other studies also indicate that late biochemical recurrence often occurs without further progression to metastatic disease and death from prostate cancer.[9 10 23] This finding can be explained by the slow progression of a prostate cancer with late biochemical recurrence and by increasing competing risks for death with ageing. According to our results there is indeed no advantage to find a biochemical recurrence more than 10 years after radical prostatectomy even in men with non-favourable histopathological characteristics unless they have a very long life expectancy.

Anxiety during follow-up is described in a variety of cancers including prostate cancer.[30] Scanxiety captures the particular distress reported by patients who are scheduled for surveillance and imaging to detect disease progression.[31] The same phenomenon is described for prostate cancer survivors who experience physical or emotional distress around the time of PSA-testing.[32] In addition, every PSA-test further diagnostic work-up and subsequent contact with a healthcare professional consume healthcare resources and it is important to balance risk and benefit with follow-up.

SPCG-4 included patients enrolled from 1989 to 1998, before the PSA era. The trial was completed before the ISUP05 consensus where some Gleason three patterns by definition shifted to Gleason four resulting in an upwards grade shift.[33] Also, most tumours were palpable, and in general more advanced than those detected by PSA-testing in more modern radical-prostatectomy cohorts. This probably explains the high proportion of ≥pT3 tumours and biochemical recurrences.

PSA-testing remains the cornerstone in follow-up after radical prostatectomy and new imaging techniques with high sensitivity to detect local recurrence and metastases are used predominantly in patients with biochemical recurrence.[2] If a biochemical recurrence does not lead to clinical progression, these imaging techniques may cause over-detection and as a corollary, over-treatment. Hence, it is profoundly important to understand if a PSA recurrence anticipates clinical progression.

## Strengths and limitations

The size of the cohort is a limitation. The cohort was too small to perform a multivariable analysis including potential confounding factors. Further, there is about 7% missing data which is a limitation. We have imputed that data to reduce bias. In a randomised trial like SPCG-4, there is a risk that the cohort is healthier than the general population which decreases the risk for death from other cause in the competing risk analysis. This might make the results less generalisable. Also, the proportion of pT3 tumours (45%) in our study is high compared with a modern cohort which affects the generalisability. However, the long and complete prospective follow-up of all relevant endpoints and an independent committee that determined cause of death strengthen the validity of our results.

## Conclusion

Our study indicates that men with favourable histopathology without biochemical recurrence at 5 years after radical prostatectomy can stop follow-up earlier than 10 years after radical prostatectomy while men with adverse pathology should continue with at least 10 years follow-up.

### Author affiliations

[1]Department of Surgical Sciences, Uppsala University, Uppsala, Sweden
[2]Clinical Effectiveness Group, Institute of Health and Society, University of Oslo, Oslo, Norway
[3]Department of Medical Epidemiology and Biostatistics, Karolinska Institutet, Stockholm, Sweden
[4]Department of Urology, Faculty of Medicine and Health, Örebro University, Örebro, Sweden
[5]Department of Oncology, Sahlgrenska Academy at the University of Gothenburg, Division of Clinical Cancer Epidemiology, Institute of Clinical Sciences, Gothenburg, Sweden
[6]School of Cancer & Pharmaceutical Sciences, King's College London, London, UK

**Acknowledgements** The authors would like to thank the Scandinavian Prostate Cancer Group (SPCG) for enabling the study.

**Contributors** The study was conceived, planned and designed by MSA, HG, LH and AB-A. H-OA, OA, J-EJ, GS, LH and AB-A acquired the data. Statistical analysis was done by MSA and HG. MSA, LH and AB-A drafted the manuscript. All authors contributed with interpretation of results, revised and reviewed the manuscript. MSA, HG, LH and AB-A had full access to the full data. The corresponding author (MSA) is guarantor of the study and had the final responsibility to submit for publication. All authors provided final approval to submit the manuscript for publication.

**Funding** This work was supported by grants from the Swedish Cancer Society (2016/466 and 2014/1275), the Swedish Research Council (2016-00177 and 2016-01293), the Swedish Prostate Cancer Foundation (grant number: N/A) and the Percy Falk Foundation (grant number: N/A) to Dr Bill-Axelson. The funders had no role in the study design or in the collection, analysis or interpretation of data, writing the report or the decision to submit the article for publication.

**Competing interests** None declared.

**Patient and public involvement** Patients and/or the public were not involved in the design, or conduct, or reporting, or dissemination plans of this research.

**Patient consent for publication** Not applicable.

**Ethics approval** This study involves human participants and was approved by Ethics review board at Örebro University Hospital (nr.251/89). Participants gave informed consent to participate in the study before taking part.

**Provenance and peer review** Not commissioned; externally peer reviewed.

**Data availability statement** Data are available upon reasonable request. Upon request to the study group (mats.ahlberg@surgsci.uu.se), data that underlie the results reported in this article will be shared, after de-identification, with researchers who provide a methodologically sound proposal to achieve the aims in the approved proposal.

**ORCID iDs**
Mats Steinholtz Ahlberg http://orcid.org/0000-0002-2404-5890
Lars Holmberg http://orcid.org/0000-0003-4417-7396

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
