## [Reviewer comments · BMJ Open]

ARTICLE DETAILS

TITLE (PROVISIONAL)	Time without PSA recurrence after radical prostatectomy as a predictor of future biochemical recurrence, metastatic disease and prostate cancer death: a prospective Scandinavian cohort study
AUTHORS	Ahlberg, Mats; Garmo, Hans; Adami, Hans-Olov; Andrén, Ove; Johansson, Jan-Erik; Steineck, Gunnar; Holmberg, Lars; Bill-Axelsson, Anna

VERSION 1 – REVIEW

REVIEWER	Mufaddal Mamawala Johns Hopkins University School of Medicine
REVIEW RETURNED	17-Nov-2021

GENERAL COMMENTS	The objective of this study was to evaluate the longer-term risks of BCR, mets and death from prostate cancer, in men without BCR within the first 5 and 10 years respectively using the SPCG-4 trial data. While the research question is important, the methodological and statistical reporting can we further refined to make this study publishable. I have following comments/concerns • It is not clear to me which statistical method was used to derive the 10, 15 and 20 year survival estimates, was a competing risk model used? If so it needs to be specified in the method or appropriately referenced.•What was the denominator for men who survived upto 5/10 years without BCR?•In the figure please provide number at risk at each of the time points in the survival curves to give an idea to the reader about the validity of the estimates. Its possible that the men at risk of event at 20 years would be very low.•Did the authors do a multivariable analysis, to see if each of the 3 risk factors, Grade Groups/positive margins/pathological stage, were independently predictive of BCR in the group? The analysis reported here is more like a univariate analyses.•What was the median follow-up of men who did not have the events of interest? That metric is more informative than median follow-up of the whole group.•Is the x-axis for all the graphs in Figure 2, “Years since RP without BCR” or just “Years since RP”? Also the x-axis on the graphs do not go up to 20 years.•The authors state: “The 20-year probabilities of metastases and
---

	death from prostate cancer in the whole cohort decreased by more than two-thirds during the first three years after radical prostatectomy without biochemical recurrence, after which the decrease flattened out” – Please provide actual estimates from the analyses. •Why weren't the estimates age-adjusted? Minor • There is a typo in figure 1, 'ranomisation' instead of 'randomisation'
--	--

REVIEWER	Zilvinas Venclovas Lithuanian University of Health Sciences
REVIEW RETURNED	20-Jan-2022

GENERAL COMMENTS	The authors of this work were willing to prove, that follow-up can be shortened for many patients after RP without increasing the risk for prostate cancer death. They investigated the long-term probabilities of PSA recurrence, metastases and prostate cancer death in patients without biochemical recurrence five and ten years after radical prostatectomy. The authors managed to demonstrate, that patients with favorable PCa features could be spared from further annual PSA testing if there is no disease recurrence in the first several years. These results are quite obvious because the literature demonstrates that patients with low-risk PCa features could be spared and could undergo active surveillance. According to NCCN guidelines, even men with intermediate-risk favorable PCa features could be spared from radical treatment. The work is well-written, the aim and conclusions are quite clear. The biggest strength of this study is a long-term follow-up after the RP. there was a comparison between different GS ($\leq 3+4$ and $>4+3$); pathological stage (pT2vs. pT3) and negative vs. positive surgical margin status. There is evidence, that lymph node invasion is also associated with poorer oncological outcomes. What are the results of your study? In table Nr. 1 the authors demonstrate that there are 7.3% of cases that are missing the final pGS, the pathological stage is missing in 7% of cases and surgical margin status in 7.3%. Could the authors explain more clearly whether you inserted these patents in the final study? in my opinion, these patients should be excluded from further statistical analysis. Authors could make a new table where the frequency of biochemical recurrence, metastasis, and death from PCa are demonstrated. Also, it would be interesting to see how many people were still alive after 5, 10, 15 and 20 years after follow-up. page 13 line 39 authors wrote that Among men with Gleason score $\leq 3+4=7$, 11 out of 111 men later received hormonal treatment for prostate cancer of whom two also underwent salvage radiotherapy. But in Table Nr. 1 it is shown that there are 174 patients with GS $\leq 3+4=7$. Could authors explain this difference? The biggest disadvantage of the work is the small size of the entire cohort.
---

VERSION 1 – AUTHOR RESPONSE

Reviewer: 1

Dr. Mufaddal Mamawala, Johns Hopkins University School of Medicine
Competing interests of Reviewer: None

Comments to the Author:

The objective of this study was to evaluate the longer-term risks of BCR, mets and death from prostate cancer, in men without BCR within the first 5 and 10 years respectively using the SPCG-4 trial data. While the research question is important, the methodological and statistical reporting can be further refined to make this study publishable. I have following comments/concerns

1) It is not clear to me which statistical method was used to derive the 10, 15- and 20-year survival estimates, was a competing risk model used? If so it needs to be specified in the method or appropriately referenced.

Answer:

In the section “Methods/Statistical analysis” we have now specified the statistical method and added a reference (ref. no. 16) (page 7, paragraph 2, line 12-13).

2) What was the denominator for men who survived up to 5/10 years without BCR?

Answer:

The denominator is “men without biochemical recurrence” and we have now specified this in the manuscript, section “Methods/Statistical analysis” (page 7, paragraph 2, line 11-12).

3) In the figure please provide number at risk at each of the time points in the survival curves to give an idea to the reader about the validity of the estimates. It’s possible that the men at risk of event at 20 years would be very low.

Answer:

We agree with the reviewer and have now provided “numbers at risk” in the probability-curves in figure 3 (former figure 2). “Numbers at risk” in this figure represent the number of patients at each timepoint who have not yet experienced biochemical recurrence and still are at risk for the event of interest.

4) Did the authors do a multivariable analysis, to see if each of the 3 risk factors, Grade Groups/positive margins/pathological stage, were independently predictive of BCR in the group? The analysis reported here is more like a univariate analysis.

Answer:

A multivariable analysis in this setting, including age-adjustment, would have required a prediction model for the analysis. The aim of the study was not to produce a validated prediction-model as the size of the cohort is rather small. To produce a validated prediction-model we would have had to divide the data-set into a “training” data-set to produce the prediction-model and a “validation” data-set to validate the model. For this, the data-set was too small. We could build a prediction-model of our data-set and validate it in another cohort and this might be an issue for future studies. We have added a section in “Discussion/Strengths and limitations” where we discuss this matter (page 18, paragraph 1, line 2-3).

5) What was the median follow-up of men who did not have the events of interest? That metric is more informative than median follow-up of the whole group.

Answer:

Altman et al. ¹ and Schemper et al.² suggest the use of reverse Kaplan-Meier estimate of potential follow-up for routine quantification of follow-up time. According to them the use of “observation time” or “censoring time” for reporting follow-up underestimates follow-up time. We have now changed in the manuscript according to those recommendations in the “Result”-section in the abstract (page 3, line 18), in section “Results/Descriptive characteristics” (page 8, paragraph 1, line 5), and specified the methodology in the “Methods/Statistical analysis” section (page 7, paragraph 2, line 15-16).

6) Is the x-axis for all the graphs in Figure 2, “Years since RP without BCR” or just “Years since RP”? Also, the x-axis on the graphs do not go up to 20 years.

Answer:

The X-axis in figure 3 (former figure 2) is “time after radical prostatectomy without biochemical recurrence”. The curves for both metastases and prostate cancer death drop to zero at 10 years on the X-axis (=10 years after radical prostatectomy without biochemical recurrence). Thus, a longer X-axis would not be informative. We have changed in the figure legend of figure 3 (former figure 2) (page 21, paragraph 3, line 8) to make this clearer.

For future biochemical recurrence we deemed it interesting to show the curves up to “15 years after RP without BCR” to demonstrate that there still is a risk for biochemical recurrence many years after radical prostatectomy.

The outcomes 10, 15 and 20 years after radical prostatectomy for men who were free from biochemical recurrence 5 years after surgery are shown in table 2.

7) The authors state: “The 20-year probabilities of metastases and death from prostate cancer in the whole cohort decreased by more than two-thirds during the first three years after radical prostatectomy without biochemical recurrence, after which the decrease flattened out” – Please provide actual estimates from the analyses.

Answer:

We have provided the estimates in the manuscript in section “Results/Outcomes 20 years after radical prostatectomy conditioned on 5 years without biochemical recurrence” (page 13, paragraph 1, line 4-6).

8) Why weren't the estimates age-adjusted?

Answer:

See answer to question no. 4.

Minor

9) There is a typo in figure 1, ‘ranomisation’ instead of ‘randomisation’

Answer:

We have corrected the typo in figure 1 accordingly.

Reviewer: 2

Dr. Zilvinas Venclovas, Lithuanian University of Health Sciences

Competing interests of Reviewer: I have no competing interests

Comments to the Author:

The authors of this work were willing to prove, that follow-up can be shortened for many patients after RP without increasing the risk for prostate cancer death. They investigated the long-term probabilities of PSA recurrence, metastases and prostate cancer death in patients without biochemical recurrence five and ten years after radical prostatectomy.

The authors managed to demonstrate, that patients with favorable PCa features could be spared from further annual PSA testing if there is no disease recurrence in the first several years. These results are quite obvious because the literature demonstrates that patients with low-risk PCa features could be spared and could undergo active surveillance. According to NCCN guidelines, even men with intermediate-risk favorable PCa features could be spared from radical treatment.

The work is well-written, the aim and conclusions are quite clear.

The biggest strength of this study is a long-term follow-up after the RP.

1) There was a comparison between different GS ($\leq 3+4$ and $\geq 4+3$); pathological stage (pT2vs. pT3) and negative vs. positive surgical margin status. There is evidence, that lymph node invasion is also associated with poorer oncological outcomes. What are the results of your study?

Answer:

In the radical-prostatectomy group in SPCG-4, surgery started with dissection of pelvic lymph nodes. If no nodal metastases were found in frozen section, a prostatectomy was carried out. In our study, only patients who had undergone radical prostatectomy were included and they had no evidence of lymph node invasion. Therefore, we cannot make any statements about oncological outcomes for patients with lymph node invasion. We have changed in the manuscript, section "Methods/Patients" to make this clearer (page 6, paragraph 1, line 10).

2) In table Nr. 1 the authors demonstrate that there are 7.3% of cases that are missing the final pGS, the pathological stage is missing in 7% of cases and surgical margin status in 7.3%. Could the authors explain more clearly whether you inserted these patents in the final study? in my opinion, these patients should be excluded from further statistical analysis.

Answer:

Missing data were handled by imputation using multiple imputation by chained equations (MICE). Imputation of missing data reduces bias in the analysis and is a common and recommended way of handling missing data.³ In our study results of data-analyses before and after imputation did not change the main result. We have changed in the manuscript to make this clearer in the section “Methods/statistical analyses” (page 7, paragraph 2, line 17-18).

3) Authors could make a new table where the frequency of biochemical recurrence, metastasis, and death from PCa are demonstrated. Also, it would be interesting to see how many people were still alive after 5, 10, 15 and 20 years after follow-up.

Answer:

We have made a new figure with cumulative incidence of biochemical recurrence metastasis and PC-death after radical prostatectomy, with “numbers at risk” (figure 2). We have also added some information about this in the manuscript (page 7, paragraph 1, line 2; page 10, paragraph 1, line 2-3) and a figure legend (page 21, paragraph 2, line 4-6).

4) Page 13 line 39 authors wrote that Among men with Gleason score $\leq 3+4=7$, 11 out of 111 men later received hormonal treatment for prostate cancer of whom two also underwent salvage radiotherapy. But in Table Nr. 1 it is shown that there are 174 patients with GS $\leq 3+4=7$. Could authors explain this difference?

Answer:

174 are the total number of men with Gleason score $\leq 3+4=7$ while 111 are the number of men with Gleason score $\leq 3+4=7$ without biochemical recurrence 5 years after radical prostatectomy. We have added a statement in the manuscript, section “Results/Outcomes 20 years after radical prostatectomy conditioned on 5 years without biochemical recurrence” to clarify this (page 12, paragraph 1, line 22-23).

5) The biggest disadvantage of the work is the small size of the entire cohort.

Answer:

We agree with the reviewer that the small size of the cohort is a limitation. This is something that we acknowledge in the discussion under “Strengths and limitations “(page 18, paragraph 1, line 1-2).

VERSION 2 – REVIEW

REVIEWER	Zilvinas Venclovas Lithuanian University of Health Sciences
REVIEW RETURNED	06-Apr-2022
GENERAL COMMENTS	Well done, congratulation